# Proximal Active Optical Sensing Operational Improvement for Research Using the CropCircle ACS-470, Implications for Measurement of Normalized Difference Vegetation Index (NDVI)

**DOI:** 10.3390/s23115044

**Published:** 2023-05-24

**Authors:** Matthew M. Conley, Alison L. Thompson, Reagan Hejl

**Affiliations:** 1U.S. Arid-Land Agricultural Research Center, U.S. Department of Agriculture, Agricultural Research Service, Maricopa, AZ 85138, USA; reagan.hejl@usda.gov; 2U.S. Department of Agriculture, Agricultural Research Service, Pullman, WA 99163, USA; alison.thompson@usda.gov

**Keywords:** active optical reflectance, NDVI, proximal phenotyping, high-throughput phenotyping

## Abstract

Active radiometric reflectance is useful to determine plant characteristics in field conditions. However, the physics of silicone diode-based sensing are temperature sensitive, where a change in temperature affects photoconductive resistance. High-throughput plant phenotyping (HTPP) is a modern approach using sensors often mounted to proximal based platforms for spatiotemporal measurements of field grown plants. Yet HTPP systems and their sensors are subject to the temperature extremes where plants are grown, and this may affect overall performance and accuracy. The purpose of this study was to characterize the only customizable proximal active reflectance sensor available for HTPP research, including a 10 °C increase in temperature during sensor warmup and in field conditions, and to suggest an operational use approach for researchers. Sensor performance was measured at 1.2 m using large titanium-dioxide white painted field normalization reference panels and the expected detector unity values as well as sensor body temperatures were recorded. The white panel reference measurements illustrated that individual filtered sensor detectors subjected to the same thermal change can behave differently. Across 361 observations of all filtered detectors before and after field collections where temperature changed by more than one degree, values changed an average of 0.24% per 1 °C. Recommendations based on years of sensor control data and plant field phenotyping agricultural research are provided to support ACS-470 researchers by using white panel normalization and sensor temperature stabilization.

## 1. Introduction

The objective perception of healthy vegetation is an important human endeavor. Past research used indices of vegetation reflectance [1,2] such as the most common Normalized Difference Vegetation Index (NDVI) to evaluate photosynthetically active plant biomass in field conditions from both remote and proximal sensing platforms [3,4,5,6,7]. NDVI normalizes the ratio between Red and Near Infrared (NIR) light, where Red light is typically absorbed by healthy vegetation and NIR light is reflected [8,9,10]. Given proper control, NDVI can resolve plant nitrogen status [11] and indirectly plant water stress [12,13,14,15].

There are different approaches to NDIV measurement, such as using the active proximal GreenSeeker [16] and in general, plant phenotyping leverages a fusion of multiple metrics like thermal and imaging to gain understanding of how plants respond to their growing environment [17]. However, due to high variability in environmental and vegetation data, reflectance sensor detector noise can become difficult to quantify absent a standard reverence.

The topic of this paper is to describe how phenotyping researchers who use the proximal ACS-470 reflectance sensor could benefit from understanding minor detector performance variance and temperature influences resolved by measurement of a white panel over time. The aim of this paper is to example several instances of ACS-470 proximal reflectance data that describe basic signal performance and possible sources of signal noise, and to offer operational suggestions for research. The novelty of this paper derives from robust experience, using two dozen ACS-470 sensors, measuring white panels for hours over years, across different conditions, and using the sensors in many experiment collections. Authors are not aware of other studies that present results informed by hours of ACS-470 white panel control data with concurrent sensor temperature measurement and offer guidance for research purposes.

This paper is structured to communicate an overview of NDVI sensing from the viewpoint of using proximal active sensing for HTPP research and to describe elements specific to the ACS-470 sensor. Section 2 speaks to the general measurement setup and approach that was conducted. Section 3 offers several examples and explanations of data collected using the ACS-470 in controlled and field conditions. Section 4 summarizes the meaning of those results and relates findings to operational recommendations. Section 5 includes limitations and future work ideas as well as a performance improvement generalization estimation.

Although NDVI is a common plant evaluation metric, any specific NDVI calculation can be physically different based on the sensor elevation, the field of view sampled, as well as the bandwidth and centering of the spectral radiometric inputs [18,19,20]. Along with other vegetative indices, many sensor products and Earth observation satellites measure NDVI. However, care can be taken when selecting or comparing between different NDVI measurements because the specific detector view position, spectral input, and bandwidth employed determines the responsive and descriptive extent that a calculated vegetation index will represent [7,21,22]. Unlike remote sensed NDVI [23,24,25,26,27,28], proximal NDVI avoids most terrain shadow, atmospheric condition, bidirectional reflectance, and anisotropy effects since the sensor is typically positioned 2 m or less distant from its target.

Active proximal NDVI sensing using the CropCircle ACS-470 is different in approach than other passive types of NDVI measurement. Passive NDVI sensors may use up looking cosine diffuser radiometers to account for the spectral intensity of the incident solar radiation [29,30] or calculate an atmospheric correction [31,32,33]. During a data collection period, solar radiation can add substantial energy to the local sensing space [34,35]. Conversely, the functional application of active NDVI sensing seeks to physically simulate and quantify the Red and NIR elements of the natural passive solar reflectivity phenomena without the inclusion of solar radiation.

ACS-470 reflectance measurement is ambient illumination independent [36]. The sensor projects an active light beam width 0.82 × height according to the operator’s manual, or a field of view approximately 32 by 6 degrees [37] and when positioned at a 1 m height, it illuminates a full optical spectrum interface footprint of ≈1.25 m × 0.25 m onto the target space which is then sampled by three onboard detectors. The consistent panchromatic LED output of the ACS-470 enables a reflectivity value of biological consequence comparable to passive reflectance sensing that is viable for research purposes [38,39,40], and the sensor also operates at oblique angles or in darkness, although only nadir views during the day are examined here.

CropCircle ACS-470 sensors are production grade active optical reflectance radiometric instruments used for proximal field vegetation assessment measurements such as NDVI [41,42]. The reflectance measurement is achieved using pulse modulated band-pass filtered light that is digitized to a raw serial numeric output [43,44,45,46]. Custom 12.5 mm filter options and a manual sensor normalization are unique to the ACS-470 product. Therefore, given an increased operational input including filter customization, the ACS-470 sensor possesses improved measurement potential for plant phenotyping research purposes compared to many other multi-spectral proximal sensors.

ACS-470 generated NDVI is typically used to indicate a nitrogen status in green plant presence within a working distance of 60 to 200 cm [47,48,49,50]. However, ACS-470 active optical NDVI can be derived in different ways. The NDVI exampled in this paper is a calculated normalized ratio using Andover Corporation custom bandpass filtered reflected color Red 670 and NIR 800 or 820 nm spectral inputs, respectively of 10 and 20 nm widths, NDVI = (NIR − Red)/(NIR + Red). Related research also uses the quality-controlled and calibration documented filters from the Andover supplier [51,52,53] which can be ordered to specification. Although the Holland Scientific supplied filters did appear to be of similar quality and performance as those from Andover, they did not include calibration datasheets. Because different 12.5 mm band-pass filters can be used in the ACS-470 sensor, and various filters may be used in other passive sensors, it is a good practice for NDVI researchers to delineate the light filter used for investigation.

Responsivity of a photoconductive radiant detector may be affected by temperature [54]. Due to physical material properties, the emission and transport of electrons across a silicone photodiode device is increasingly resisted as temperature increases [55,56]. Although performance of the sensor’s electronic components is expected to drive the fundamental quality of the sensing signal, the specific ACS-470 proprietary electronic components and possible internal sensor thermal mitigation signal adjustments are unknown. The influence of temperature is not addressed in the operator’s manual; however, the optical radiometer product could theoretically exhibit a detector value drift due to a change in thermal status. Moreover, the option of user selected band-pass optical filters placed in front of the individual detectors could modify the sensor performance or otherwise change the sensitivity of an individual detector response.

## 2. Materials and Methods

Sensors and data logging involved two dozen CropCircle ACS-470 reflectance sensors and four corresponding GeoSCOUT X, two SC-1 normalization boxes, and associated control software (Holland Scientific Inc., Lincoln, NE, USA) and connective cabling including 12-volt power supply. Sensors were characterized for use in field phenotyping research by recording the sensor temperature with nadir view reflectance at standardized distances. Sensors were equipped with Type-T thermocouples (Omega Engineering Inc., Norwalk, CT, USA) to determine continual thermal status and later insulation was added to the sensor body for thermal stabilization. The thermocouples (TC) were surface mounted, typically on top of the sensor bodies laterally between the emitting LEDs and radiance detectors. This position was chosen as a representative sensor thermal status location, visualized using T650sc thermal infrared camera imaging (Teledyne FLIR LLC, Wilsonville, OR, USA) (Figure 1). TC temperature recording is a common method [57,58] used on sensor bodies for control [59,60,61]. The TC junction tips were factory supplied or sealed using viny liquid electrical coating or electrical heat shrink tubing. TC junctions placed on the sensor housing outside surface were covered with Reflectix (Reflectix Inc., Markleville, IN, USA) mylar bubble wrap insulation attached with adhesive aluminum foil metal duct tape. TC junctions were also attached to a sensor internal electronic board for testing. TC wires connected to either a CR1000 or CR3000 Campbell Scientific data acquisition system. (Campbell Scientific, Logan, UT, USA) running CRBasic control software which provided the temperature corrected analog to digital conversion logged at 1–5 Hz.

Sensor reflectance was standardized using a white normalization reference. ACS-470 sensor detector signal performances were resolved using white painted field panels placed at a standard distance of 1.2 m below the leveled sensors in nadir view. The sensor raw detector signal light intensity of $5326.4\times \mathrm{distance}\mathrm{in}{\mathrm{cm}}^{-1.762}$ was measured in a laboratory by moving the active sensor (SN#124) towards and away from the white panel at 5 cm increments (75 cm to 120 cm) using a #260061 Wesco mobile air lift table (Westco, Pittsburgh, PA, USA) (Figure 2a). To measure eight sensors simultaneously, a 4.88 m × 0.92 m × 0.02 m (l × w × h) wood framed plywood hinged panel setup (Appendix A) was painted white with high titanium-dioxide content (Behr Ultra-Pure White #1850, Behr, Santa Ana, CA, USA). Titanium-dioxide (TiO_2_) was used for all white panels and is known to have a flat refection (98–100% of solar irradiance) across the optical spectrum [62,63,64]. A 2.44 m × 1.22 m × 0.019 m (l × w × h) white painted plywood panel was used to measure two sensors concurrently (Appendix A). The 1.2 m distant white panel approach was sufficient to achieve sensor reflectance values equivalent with those produced by the Holland Scientific SC-1 single sensor field normalization box, which notably is required to communicate with the sensor to achieve detector normalization (Figure 2b). For this study, filters were also removed, and the unfiltered reflectance normalized and recorded to determine basic sensor performance absent the band-pass filter influence.

Sensors were tested using an environment-controlled room. To determine the effects of sensor warm-up and changing ambient temperatures on reflectance values in a controlled environment, the ACS-470 sensors with attached thermocouples and data loggers were measured in an Elliott-Williams Company temperature-controlled calibration room (Elliott-Williams Company Inc., Indianapolis, IN, USA). The room was 3 m × 3 m × 5.25 m in size and equipped with a Marley Engineered Products ST Series commercial slope-top convector heating unit (Marley Engineered Products, Bennettsville, SC, USA) with 1500 watt metal sheath element and A421 PENN Johnson Controls electronic temperature controller (Johnson Controls, Milwaukee, WI, USA), as well as a Russell extra low-profile unit Ceiling-Temp 4 motor 20-amp 14,000 BTU condensing forced air cooler (Russell, Scottsboro, AL, USA) with a Ranco Electronic Temperature Control ETC-111000-000 (ETC Supply, Delphos, OH, USA). The temperature stability and available 1 to 50 °C ambient range for testing the equipment was maintained by the temperature control room thermal envelope and its heating and cooling powered equipment. Temperature measurements of the air and sensor bodies were independently recorded. Testing was also partially conducted in a laboratory with somewhat stable ambient conditions (≈23 °C) at the US Arid-Land Agricultural Research Center (ALARC) in Maricopa, Arizona. One sensor was temperature stress tested using a Hobbico HCAR7000 1000-watt heat gun (Hobbico, Champaign, IL, USA).

Sensors were measured in field conditions as part of a phenotyping platform. After sensors were characterized in controlled conditions, they were mounted to a retrofitted LEE Avenger (AvengerPro) high-clearance tractor (LeeAgra Inc., Lubbock, TX, USA) with a modified front boom as described by [65], and to the in-house developed “Wolverine” proximal sensing cart, 1 m above an average crop canopy height and in nadir view for use in experiment data collections [66]. Field trials were conducted at the Maricopa Agricultural Center (MAC) in Maricopa, Arizona, USA (33.079° N, 111.977° W, 360 m above sea level) between years 2012–2022 on a variety of row-crops including cotton, wheat, barley, sorghum, soybean, camelina, brassica and guayule. The data presented in this paper was collected from upland cotton breeding trials conducted between years 2015–2022 production seasons (April–October). The field trials were arranged in a (0–1) alpha lattice design with 10.60 m × 1.02 m experimental plots. For trials after 2015, additional solar and thermal sensor isolation was provided by wrapping metalized polyethylene bubble insulation (Reflectix) around the sensor bodies. Data recording began inside an open door garage for the Avenger tractor, or outdoors in full sun for the Wolverine platform, with sensors leveled and 1.2 m distant to measure the white panel reference over at least a one-hour warm-up period (typically three hours, and sometimes more when time allowed) and ended with a cool down period after the field data collection with sensors still recording and returned to the white panel as described by [65]. Sensor detectors were normalized to their white panel 1.0 unity value after warmup and before field collection using the SC-1 device connected to a PC.

By measuring each detector after normalization and just before and after field collections, any difference from unity could be quantified. The difference for the pre, or post, the average, or a linear interpolation between pre and post unity offsets was available to adjust raw detector values for any field collection, as determined on an experimental basis by the principal investigator.

The Avenger and Wolverine platforms concurrently recorded air temperature and relative humidity (HCS2S3, Rotronic AG, Grindelstrasse Bassersdorf, Switzerland), solar radiation (SP-110, Apogee Instruments, Inc., Logan, UT, USA), as well as the temperatures of the ACS-470 sensor bodies (TC), in addition to other phenotyping metrics. Environmental conditions were also recorded by an Arizona Meteorological Network (AZMET) [67] weather station located 0.8 km away from the experimental plots. The weather station measured hourly air temperature, relative humidity, photosynthetically active radiation (PAR), wind speed and direction, and several other metrics. Sensor temperature effects from sun and wind were evaluated in field conditions. All sensor temperature stabilization tests were conducted after the sensors had undergone a three-hour warm-up period and achieved initial thermally stability.

Data handling and statistical analysis consisted of comma separated data tables that were processed and charted using Excel version 2212 (Microsoft Corporation, Redmond Washington, DC, USA). Linear regression models and descriptive statistics were applied to reflectance and temperature data. Additional charting and statistical analysis were performed using JMP (build 15.2.0 SAS Institute Inc., Cary, NC, USA).

## 3. Results

The results presented include several data examples. The ACS-470 raw unfiltered detector temperatures and signals for one sensor are shown during an initial warmup in the laboratory to illustrate a typical sensor operational warming (Figure 3). Four sensor temperatures and detector traces are shown when in a cooling condition inside a temperature control room to example a change in ambient temperature (Figure 4). Sensor detector values with unfiltered, Red and NIR filter influence are presented for one sensor to parameterize a basic signal behavior (Table 1). Then HTPP field experiment data is provided to show how sun (Figure 5) and wind (Figure 6) can influence sensor temperature and how physical insulation around a sensor can help stabilize its thermal condition (Figure 7). The sensor signal for Red and NIR filtered detectors is exampled from past experiment operations to show individual detector signatures during warmup (Figure 8), and to relate how temperature correlates with detector values (Figure 9). Finally, multi-year control data results from HTPP experimentation are presented to quantify a case example of detector change before and after field data collections (Figure 10 and Figure 11).

Sensor temperature influence on unfiltered detector signals

Sensor thermal status and unfiltered detector values change during a three-hour warmup period while in a stable environment. Measurement of the uninsulated external housing surface on the top middle of the sensor showed 8 °C increase above ambient temperature after one hour of warming in a laboratory environment (Figure 2). This differential increased to 10 degrees °C above ambient air temperature after three hours. A three hour logarithmic sensor warmup curve can be described (R^2^ = 0.9791) as temperature increase in °C = 3.6217 × log(time in minutes) + 12.517, when the ambient temperature was 22.4 °C. Concurrently, the unfiltered detector white panel nadir view measurements changed with the sensor thermal status (Figure 3).

Temperature influence on filtered detector signals

Fully warmed sensor detector values were sensitive to changes in ambient temperature when measured in the controlled environment room. Sensor reflectance values were affected as the temperature in the control room was changed. Sensors were normalized at 10 °C then allowed to increase in temperature to 33 °C. By using forced air cooling to decrease the ambient room temperature from 26 to 2 °C, the sensor body temperatures decreased and consequently so did the reflectance values of center detectors from four different sensors. (Figure 4).

Filter influence

Band-pass filters influence the sensor reflectance signal and how it responds to temperature change. The individual optical band-pass filter employed defines the amplitude and raw signal variance measured for each detector. Testing the ACS-470 sensor SN#124 in the laboratory environment positioned 1.2 m horizontal and level to a white painted reference panel, and normalized to unity, when at an internal 20 °C above ambient fully warmed operating temperature (after three hours of operation), for the unfiltered, the filtered Red 670, and the filtered NIR 800 measurements, showed different point-to-point variances across 100 samples recorded at 1 Hz. The unfiltered raw signal was the least variant both in total, and across the three detectors, with values of 2.7 × 10^−9^, 2.4 × 10^−9^ and 2.1 × 10^−9^, for the R1, R2 and R3 detectors respectively (detectors are typically numbered left to right if the sensor is turned upside down to face up at an observer). However, when three Red 670 nm filters were placed in front of the detectors, the variance increased to 5.5 × 10^−6^, 8.6 × 10^−6^ and 4.8 × 10^−6^, for R1, R2 and R3 detectors respectively. Use of the NIR 800 nm filters resulted in the highest variance. Repeating the test and installing NIR 800 nm filters, normalizing values to 1.0 and measuring another 100 samples after the sensor had fully thermal stabilized showed a variance of 1.6 × 10^−3^, 4.9 × 10^−4^ and 1.5 × 10^−3^ for the R1, R2 and R3 detectors respectively (Table 1).

Band-pass filters greatly reduce the magnitude of the raw unfiltered reflectance signal received by the ACS-470 sensor and determine the sensor signal final behavior. After normalizing an ACS-470 sensor SN#124 without filters, and then installing three Red 670 nm filters, results showed a raw signal reduction of 95.5% (Table 1). Likewise, the NIR 800 filters reduced the raw unfiled values by 99.7%. Therefore, because the band-pass filters only allow a small amount of the reflected radiation to enter the sensor detector, and the detector is likely not uniformly sensitive across a working range of 350 to 850 nm, it is suggested that the performance characteristics of the filter drive most of the filtered detector character differences. For example, when the sensor was normalized with the three Red 670 nm filters, brought to thermal equilibrium with the environment of the laboratory overnight, and then measured during the first hour of initial sensor warmup of 20 °C, detectors drifted 0.05%, −0.01%, and 0.01% per °C increase respectively for the R1, R2, and R3 detectors. However, when NIR 800 nm sensors were installed to the sensor and the sensor normalized at its full operating temperature, then brought to thermal equilibrium in the laboratory environment overnight and again measured for the first hour of the initial warmup period, results showed increased individual detector drifts of −0.33%, −0.19%, and −0.95% per °C across a 20 °C temperature increase.

Influence of Environment in Field ConditionsSun and Wind

The ACS-470 sensor thermal status can change when operated in the field environment due to sun and wind influence, but these effects are mitigated when the sensor is insulated. Environmental insolation can increase the ACS-470 sensor body temperature during operation. Sunlight that directly hits the sensor housing serves to increase sensor temperature beyond what the active illumination warming alone would induce. Likewise, solar shading of the sensor during operation serves to decrease sensor temperature relative to the sun-lit operational environment instance (Figure 5).

Ambient air temperature can influence the thermal status of the ACS-470 sensor and this influence increases with air velocity. Likewise, an impinging cool wind decreases the ACS-470 sensor body temperature due to convection (Figure 6). Hotter than sensor air will cause an inverse effect, but typically air that is cooler than the fully warmed sensor body temperature is encountered in a field data collection operation. Moving air is more influential on sensor temperature than is still air. The larger the temperature differential between the sensor and the ambient atmosphere, the more air temperature will affect sensor temperature status.

  2.Insulation Mitigation

Physical insulation around the sensor body increased sensor temperature and mitigated environmental thermal fluctuation. Short duration solar impingement and gaseous cold convection thermal effects were decreased by providing one or more layers of insulation material around the ACS-470 sensor bodies, while excepting the areas of light emission and measurement (Figure 7). Adding two separated layers of the 0.64 cm mylar Reflectix bubble wrap (or 3.8 cm of DuPont Great Stuff polyurethane foam insulation, DuPont, Mississauga, Ontario) around the sensor, empowered the sensor self-warming effect to increase the sensor body 20 °C above ambient air when in a still condition. However, the rugged ACS-470 sensor design can handle high temperatures during operation. To verify heat tolerance, sensor SN# 124 with Red filters installed was stress tested using the heat gun, where thermocouples affixed to the electronics board inside the sensor measured over 100 °C, yet the filtered sensor did not error nor was performance subsequently damaged other than a positive average 0.043 per °C raw detector value drift expected for that sensor setup.

Sensor signal response outdoorsWarmup before field measurement

In field experimental practice, sensor reflectance values changed during initial warmup periods and due to change in outdoor ambient temperature. Once filters are applied to sensor detectors, the signature of data measured from white panels relative to changing temperatures was not consistent between one detector and another (Figure 8). Most detectors increased in value with increasing temperature, however some decreased, and there were instances of initial value drift reversal during a consistent warmup period (Appendix B, Table A1), and less change in the post collection period (Appendix B, Table A2). Therefore, each sensor showed its own performance character as a function of the sensor serial number, the individual of the three detectors onboard a sensor, and most importantly the band-pass filter applied. This cumulative effect was largely consistent across repeated measures of the same detector setup.

  2.Reflectance and temperature

Sensor detector influence is correlated with temperature status. Two insulated sensors mounted to the “Wolverine” proximal sensing cart were placed outside over the white panel reference and allowed to warm in full sun prior to a field data collection while their body temperatures and reflectance values were recorded (Appendix A). Results show the degree to which each detector response is associated with the sensor thermal status (Figure 9 and Appendix A).

  3.Pre and post field collection reference measurements

Measuring a white panel before and after field operation illustrated sensor performance and filter influence. The white panel measurement resolved sensor detector post-normalization offsets and signal drift that occurred during a field collection period (Figure 10 and Appendix A). Field normalization of the detectors to unity was typically achieved within ± 0.01 (or 1%) before a field collection. If detectors were found to be offset ± 0.02 or more from unity, a second normalization could decrease the detector offset to between 0 and 0.01. However, achieving a normalization offset less than 0.01 usually did not occur with several additional normalizations. Therefore, detector offsets of plus or minus 1% from unity after normalization were considered functionally optimized.

  4.Detector changes during field measurements

ACS-470 detector median drift was 0.0029 (standard deviation 0.074, standard error 0.0037, ±0.0073 confidence interval at 0.95) for 400 observations between 2018 to 2021 of the Red and NIR filtered detectors, or 0.28% per °C (data in Figure 11 and included in Appendix A), measured from sensor thermal stabilization and unity normalization before to after field collections. However, other individual detector drifts were different depending on the final detector configuration and specific recording event (Appendix B, Table A1 and Table A2 examples).

## 4. Discussion

After conducting 9 years of field investigation since 2013 involving two dozen ACS-470 sensors, authors consider the Holland Scientific ACS-470 CropCircle a customizable product that is illumination and angle independent and that provides reliable performance in field conditions. It supports nitrogen management as intended but can also be custom managed to further support research activity through individual detector characterization which includes specific filter selection, applying a reference normalization, and control of thermal influence [68]. An implication for research purposes is that previous ACS-470 reflectance data has been corrected using protocol which involved sensor body temperature stabilization, field normalization, and detector performance tracking over time [69] and raw datasets [70,71,72]. Authors are not aware of other experimental usages where ACS-470 detector signals were reported to be corrected in this way.

The ACS-470 has proven resistant to harsh environmental variables such as high temperatures, blowing dust, and physical vibration. As intended by the manufacturer the sensor can supply prolonged field data collections like NDVI sensing for crop production nitrogen management. However, a key finding is that the active sensor properties, custom filter implementation option, and manual sensor normalization function using white panels make the ACS-470 a unique technology option able to support research grade proximal multi-spectral field data collections to include NDVI, or other vegetation indices, which could support crop breeding or crop simulation modeling.

It is important to allot 30 to 180 min of powered ACS-470 sensor warmup time (temperature increase in °C at 23 ambient = 4.6224 × log(time in minutes) + 4.3004) before research measurement to let the LED active lighting induced sensor body full temperature increase to occur, and thereby effect a more stable thermal sensor electronics operation condition. Also, the sensor field of view and the geometric sensitivity (percent of signal reduced = $0.00001\times {\mathrm{cm}}^{3}-0.0005\times {\mathrm{cm}}^{2}+0.0098\times \mathrm{cm}-0.004$, and 0.0003 × cm^2^ − 0.0114 × cm + 0.1066 for the length and width respectively) across that field directly determines the reflectance result beyond the white panel. This is mentioned because signal geometric response area is suggested to be determined relative to the plant and soil space [73].

White panel usage supports consistent measurement across long periods to characterize sensor warmup as well as basic detector performance. The white panel approach allows documentation of detector point-to-point variance and temperature effects on the signal by resolving filtered detector reflectance across different ambient and sensor body temperatures while in view of the same reference target. The use of white panels also supports control data for the purpose of data quality, which could extend over months or years of individual sensor use in field experimentation. NDVI reflectance measurement of soil and plants is too variable to determine minor detector drift effect with temperature or to resolve a change in sensor performance over long periods of time. Likewise, the basic comparison parameters of the detector signal such as the baseline data variance point-to-point are unresolvable when measuring only soil and plant targets. Using the white panel approach supplied a control target which revealed the basic signal noise for individual sensor detectors and resolved the signal modifying effects that individual band-pass filters introduce.

Parameterizing individual filtered detectors allows a final directed placement of the filters to optimize the sensor setup by pairing most important reflection metrics with most consistent performance filtered detectors, and to support generation of the vegetation index using detector responses which agree (may drift in the same direction due to temperature change). Authors are unaware of other work describing the specific characterization and direction of filter placement in ACS-470 sensors. Finally, by using white panel measurement before and after every field collection (along with the standard normalization routine), an adjustment of raw values based on the actual sensor signal status before, and any changes that may occur during a field collection event is possible. This approach may support an improved data result for research purpose [74].

The increased operator input process for ACS-470 research function showed that white reference panel measurement was instrumental in data quality control. The procedure steps involve the following. First the sensor was made ready for field measurement by observing the warmup period to signal stabilization. Then the field normalization was performed to set each detector output to 1.0 when positioned nominally 120 cm away from and in nadir view of the white reference panel. This also included setting the sensor output to 0.0 with the detectors and LED emitter covered by an opaque foam or with folded cloth. Next, to parameterize potential detector drift during the field data collection period, the sensors were brought back to the original white panel position after field collections were finished and measurements taken again. Often, the post field collection white panel measurement showed that sensor detectors were no longer reading the previously set unity value. Instead, they could be a one or more percent offset, where different detectors behaved differently. The amount of offset was associated with environmentally induced thermal change, but this did not explain all the measured difference. Therefore, it is suggested that a correction to the raw values of individual detectors can best be made when at minimum pre and post field collection white panel standard normalization measurements are employed.

Knowing a generalized thermal drift allowance for each detector in use offers a potential data correction based on temperature. Although a temperature measurement of the electronics board can better represent the actual detector thermal environment, this is a difficult measurement to take because it would require opening the sensor housing and compromising the environmental seal of the unit. Therefore, measuring the sensor temperature status on the top of the sensor is much easier to achieve, and when insulation is placed over the sensor surface temperature measurement area, the approach is adequate to estimate the logarithmically related internal sensor thermal status. Lastly, once the sensor has come to full warmup thermal stability, the ambient air temperature change can be used to loosely approximate a thermal status of the sensor body, where lack of wind and use of insulation around the sensor improves this estimation. Applying insulation around an ACS-470 sensor prior to operation and consequently operating the sensor at a 10 to 20 °C higher temperature, can aid sensor thermal condition stability by mitigating heat loading from direct solar radiation, warming ambient air, or cool breeze convection without harming sensor signal quality. However, ambient and sensor body temperature alone did not explain all the differences measured in white panel values pre and post field data collections. There were instances when temperature was within 1 °C from before to after the field collection, yet detectors changed by 2% or more (Figure 11). Therefore, it is the white panel measurement that is suggested to be used as the basis of any signal status determination and possible correction (Figure 10), where the temperature change is used only as a proxy when no white panel reference data is possible.

Drift in a raw detector value affects an NDVI calculation. If the two detectors which contribute to the NDVI calculation were to exhibit the same thermal drift influence and shift value with temperature to the same degree, then although the raw signal would change with temperature, the NDVI calculation would not. However, if the detectors drifted in the same direction with temperature change but to different degrees, which is usually the case, then the difference between the two detector drifts would influence NDIV in a small way. For example, if the Red detector drifted by 0.4% per °C while the NIR detector drifted in the same direction by 0.2%, NDVI would be influenced by 0.1% per °C. Moreover, if the detectors were to drift in opposite directions with temperature change, then the NDVI bias would be compounded. If the Red detector were to drift by 0.3% per °C while the NIR detector drifted by −0.3%, this would result in an NDVI change of 0.3% per °C. Because each ACS-470 sensor, detector and chosen filter employed determine the actual thermal induced value drift encountered during measurement, it is important to characterize specific detector setup bias to understand any likely thermal effect on NDVI calculation.

The practical application of measuring the ACS-470 detector values on a white reference panel before and after a field data collection is to allow characterization of each individual detector performance and enable a possible data correction. This could improve subsequent active optical reflectance analysis.

A summary list of the improvement techniques is provided.

Sensor thermal status measurement applicationSensor insulation applicationIndividual filtered detector characterizationSensor warmup before field collectionSensor unity white panel normalizationPre-field collection white panel measurementPost-field collection white panel measurementPossible data correction using pre/post unity offsets

## 5. Conclusions

CropCircle ACS-470 multi-spectral reflectance sensors perform well in field conditions and support nitrogen management. The active optical ACS-470 sensors are unique to allow manual normalization plus use of custom frequency 12.5 mm band-pass filters where they enable NDVI to sense green biomass or generate other optical vegetation indices. However, the sensor detector band-pass filter used drives resultant small temperature influenced signal changes. Understanding specific filtered detector performance enables mitigation or correction. The objectives of this study were to evaluate sensor performance across several experiment configurations and environments and make operational recommendations that could improve data quality for research. Although the sensors achieve their intended purpose when used as directed by the manufacturer, additional functionality for purposes of agricultural research are possible by measuring individual detector response, directing the custom filter option, and utilizing the expanded user protocol involving (TiO_2_ painted) white normalization reference panels and sensor temperature tracking or control.

Regarding future work, active thermal control could be an additional way to increase the research potential of the ACS-470 sensor. The solid state bi-metal junction electric current induced Peltier heat pump is well described and commonly used in electronics [75,76,77,78]. Application of Thermoelectric cooling (TEC) could hold an ACS-470 sensor at an optimized temperature setpoint. Although not tested for this paper, it is theorized based on applications supporting other radiometric sensing, that the application of several 12-volt 60-watt TEC chips could stabilize ACS-470 sensor temperature and provide a more optimal sensor energetic status for field data collections. Although additional work is warranted to prove this concept, TEC was successful to prolong Nikon camera field operations for field phenotyping data collections in 2019 [79], and to stabilize the thermal operating environment supporting an ASD FieldSpec3 measurement cotton in 2022 [80].

Limitations to defining an overall quality of the ACS-470 reflectance signal for research include an underlying detector/filter/sensor/temperature/target “stack” induced noise in the final optical reflectance, and that differences occur between detectors at different times and in different environments. There were also few instances of anomalous transient data points. These make difficult the determination of a clear statistical significance across all cases. Additionally, although a single point of basic signal performance can be resolved by measuring a white panel, anytime the sensor is in-field measuring unknown targets, the quality of that data is less known. Authors suggest that bracketing a field collection with a pre and post white panel reference measurement is sufficient to characterize sensor performance for that day of collection, however, measuring a white panel more frequently in the field between transects may offer further control. It is also difficult to directly compare the ACS-470 performance with other reflectance products due to its proprietary unique technology with customizable nature.

As a generalized guidance, the enhanced protocol for research described in this paper is expected to improve raw ACS-470 detector signal quality a median of 0.41% per °C change (standard deviation 3.13%, standard error 0.13%, confidence interval of ±0.26% at 0.95). This estimation is informed by 522 observations measuring all detectors of four sensors after normalization and again after field collection 2018 to 2021 (Appendix A). However, it is important for researchers to characterize every sensor detector used in each research collection over time using a white reference panel to determine their actual signal correction potential.

The USDA is an equal opportunity provider and employer. Mention of a trade names or commercial products in this publication is solely for the purpose of providing specific information and does not imply recommendation or endorsement by any part herein.

## Figures and Tables

**Figure 1 sensors-23-05044-f001:**
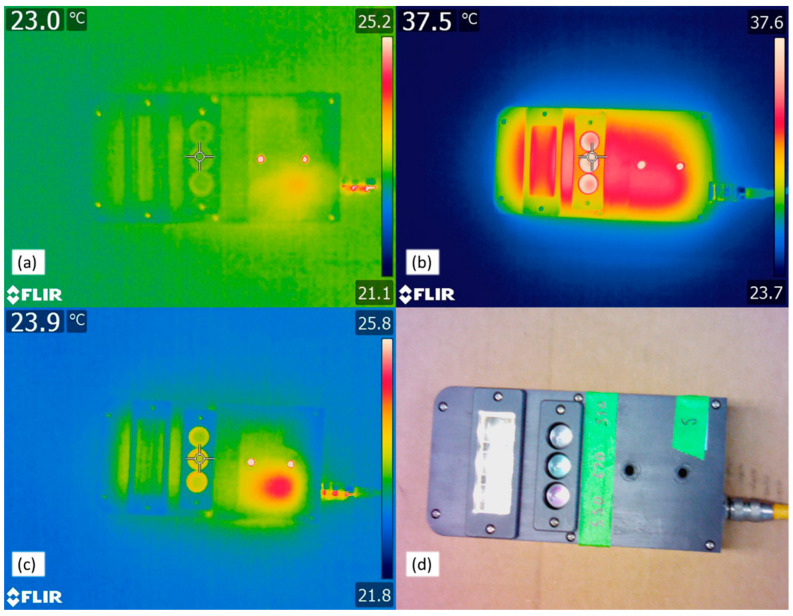
FLIR thermal images show an ACS-470 warming and an optical image of the sensor (**a**) one minute after electrification; (**b**) ten minutes after energizing; (**c**) after one hour of operation with thermal status more normalized across the sensor housing; and (**d**) a color image of the active sensor bottom face.

**Figure 2 sensors-23-05044-f002:**
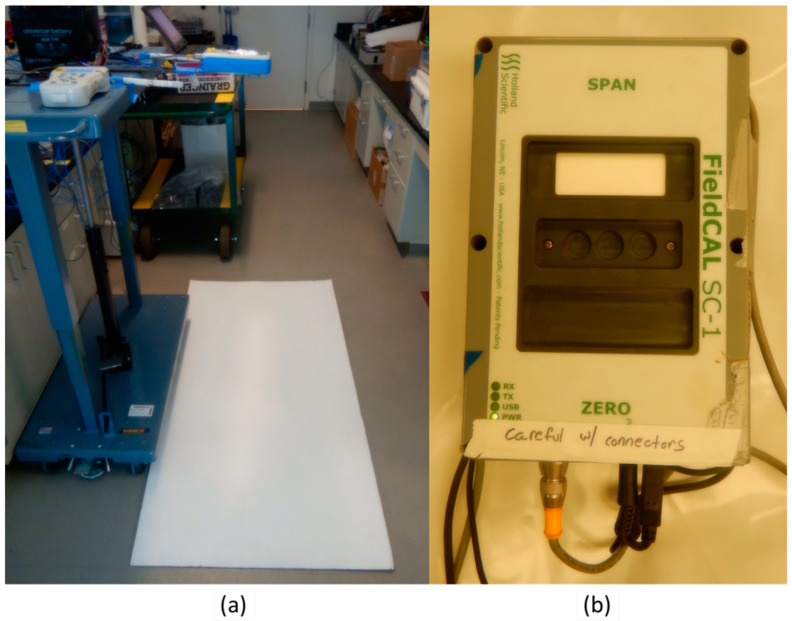
(**a**) Test sensor SN#124 recorded by GeoSCOUT X, with sensor internal and external thermocouples and ambient air sensor, in the lab space 1.2 m above a white reference panel, and (**b**) the SC-1 hardware communications and normalization box.

**Figure 3 sensors-23-05044-f003:**
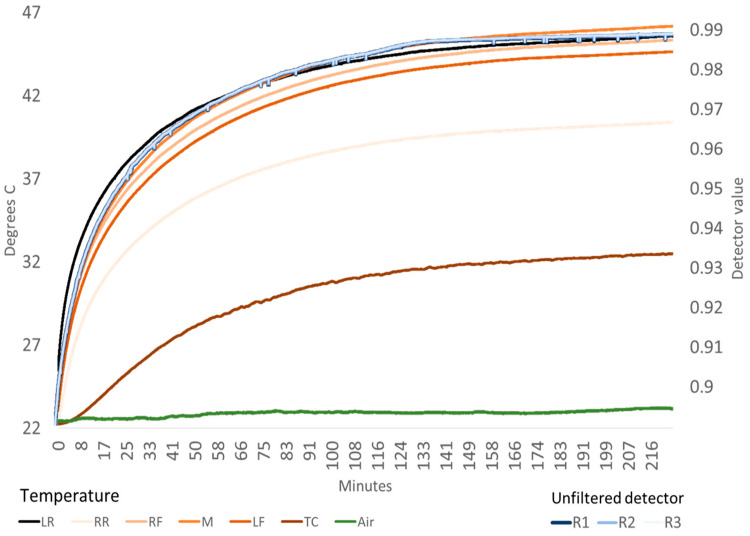
Chart of sensor SN#124 warmup temperature curves (in C, left axis) over 224 min in a controlled environment including ambient air (Air), a single external (TC = Top Center) and five internal thermocouples attached (labeled with sensor face up, LR = Left Rear, RR = Right Rear, RF = Right Front, M = Middle, and LF = Left Front). The raw unfiltered normalized detectors show change in their reflectance values (R1, R2 and R3 labeled left to right with sensor face up, right axis).

**Figure 4 sensors-23-05044-f004:**
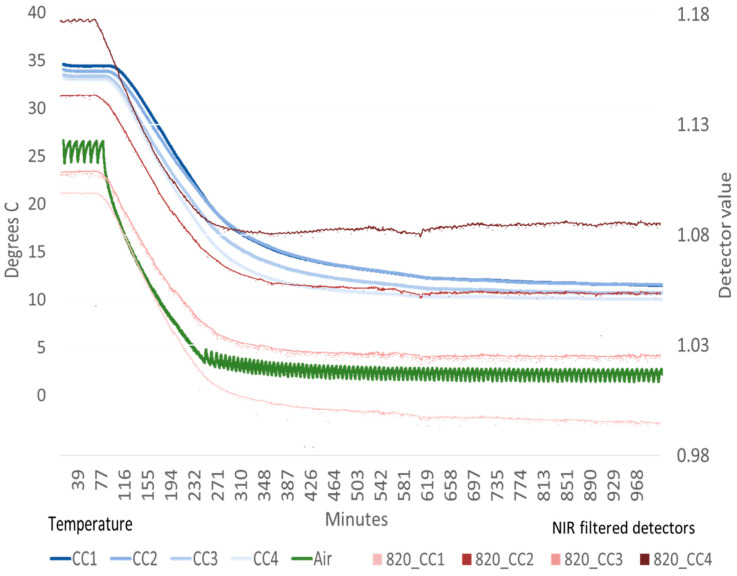
Four sensor body temperatures (CC1, CC2, CC3 and CC4), measured from top middle external sensor body thermocouples, and ambient air temperature (Air) in °C (left axis), show the sensor cooling effect measured in a temperature control room with active cooling. The reflectance values of four normalized NIR filtered (820) center (R2) detector channels (right axis) show change in values that track with the sensor temperatures (sensors CC1 = SN#306, CC2 = SN#345, CC3 = SN#145, CC4 = SN#267).

**Figure 5 sensors-23-05044-f005:**
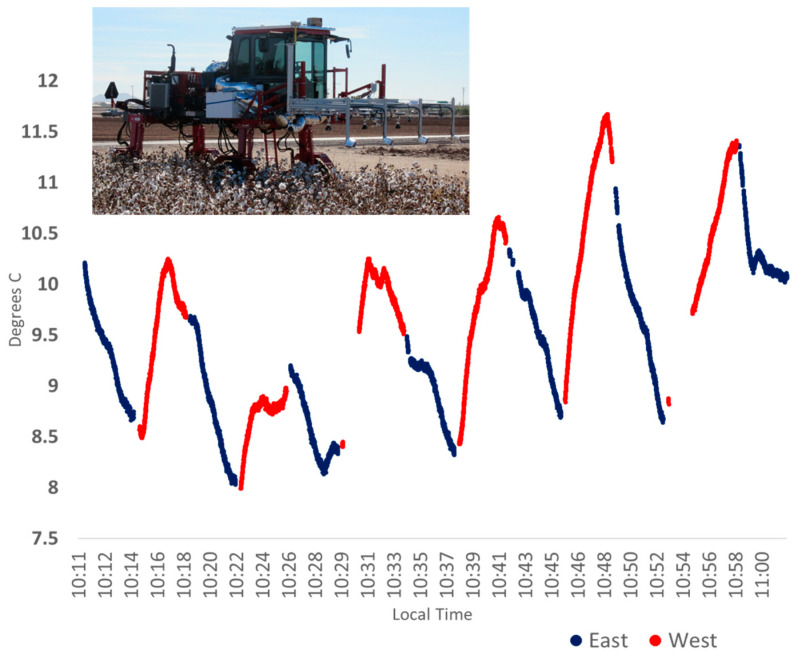
A chart of the uninsulated sensor SN# 267 (mounted on the Avenger platform 18 November 2015, as shown in image insert) body temperature difference from air in °C, changing with the sensor orientation while in the field environment moving in and out of direct sunlight. Changing rig travel orientation relative to the sun during the field data collection operation caused the sensor to experience more sun when traveling East, and more shade when traveling West. Ambient air temperature 11.6 to 15 °C.

**Figure 6 sensors-23-05044-f006:**
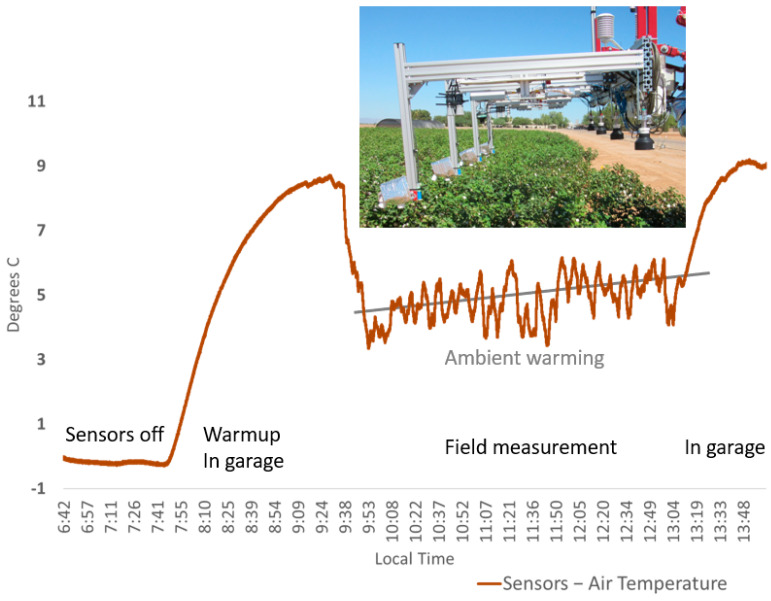
Chart of eight sensors (mounted on Avenger platform 23 September 2015 as shown in image insert) average temperature difference from air in °C and cooling with wind during a field measurement. The sensors without external insulation were fully warmed inside a garage (37 °C) then taken into the field where they encountered a breeze of cool air which decreased the sensor body temperatures during the field collection period (air averaged 27.3 °C and moved 1.48 m^−1^ (wind reported by AzMet Maricopa, year 2015, DOY 266, hours 9 to 13, https://ag.arizona.edu/azmet/data/0615rh.txt, accessed on 24 September 2015)). A typical sensor warming trend occurred during the field data collection due to ambient mid-day air warming, and then additional sensor warming is evident once the sensor is taken back into the still air of the garage.

**Figure 7 sensors-23-05044-f007:**
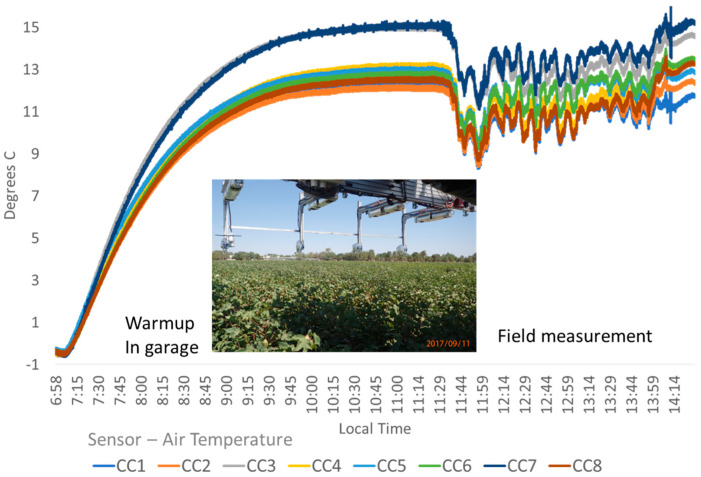
Chart of sensor temperature stabilized with insulated bodies. Adding two layers of mylar bubble wrap insulation to the sensor bodies reduced the solar and convective thermal influences of the environment. The chart shows the temperature spread and difference in °C of eight sensors (mounted to the Avenger platform 11 September 2017 as shown in the image insert) from the ambient air measured indoors during a warmup period and through a field data collection (CC1 = SN#335, CC2 = SN#264, CC3 = SN#303, CC4 = SN#256, CC5 = SN#267, CC6 = SN#333, CC7 = SN#301, CC8 = SN#217). Ambient air temperature was 38 to 41 °C.

**Figure 8 sensors-23-05044-f008:**
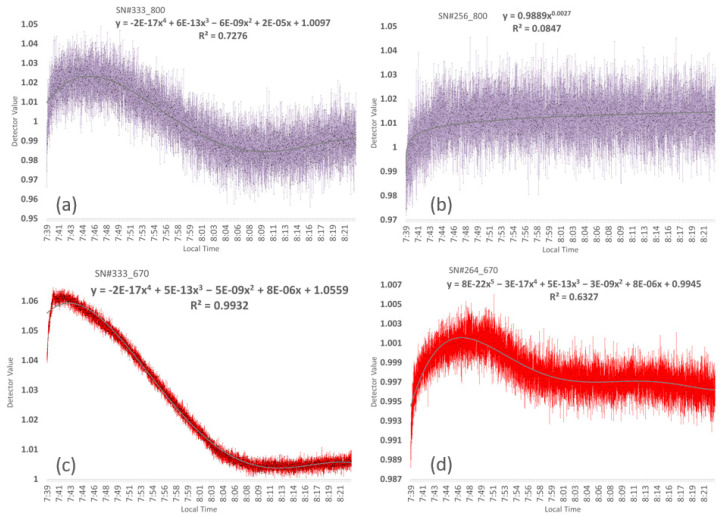
Panel of different detector performance signatures during the first 44 min of an initial warmup period before an Avenger platform field data collection on 7 August 2017. The different filtered detectors were measured in the shade of an open garage above a white panel as the sensors warmed 7 °C (31.4 to 38.8) while the ambient air warmed 1 °C (32 to 33). (**a**) SN#333 with NIR 800, (**b**) SN#256 with NIR 800, (**c**) SN#333 with Red 670, and (**d**) SN#264 with Red 670.

**Figure 9 sensors-23-05044-f009:**
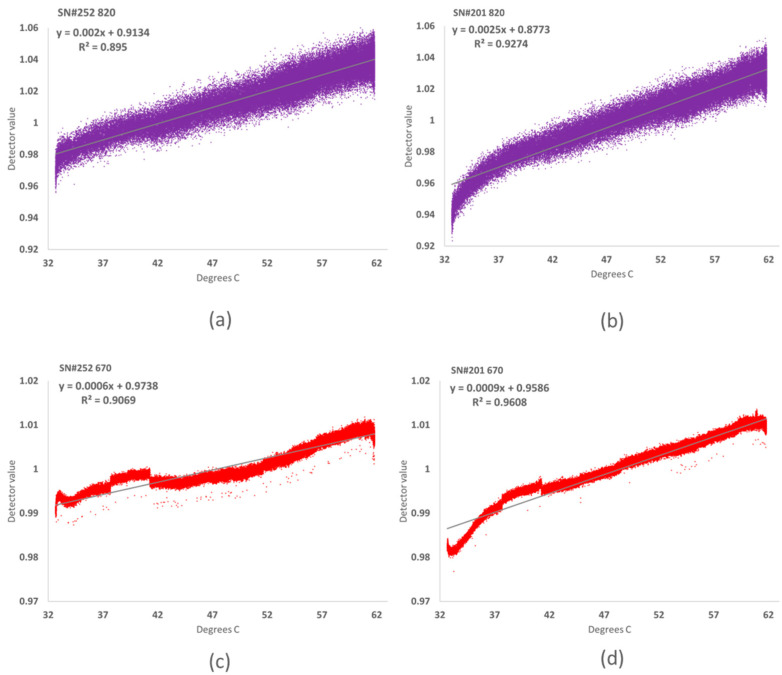
A panel of four XY scatter charts show a correlation with temperature from white panel reference data for sensors mounted to the “Wolverine” proximal sensing cart and warming outdoors in the sun before a field data collection event on 3 August 2021 ambient temperature raising from 32 to 39 °C. (**a**) SN#252 NIR 820, (**b**) SN#201 NIR 820, (**c**) SN#252 Red 670, and (**d**) SN#201 Red 670.

**Figure 10 sensors-23-05044-f010:**
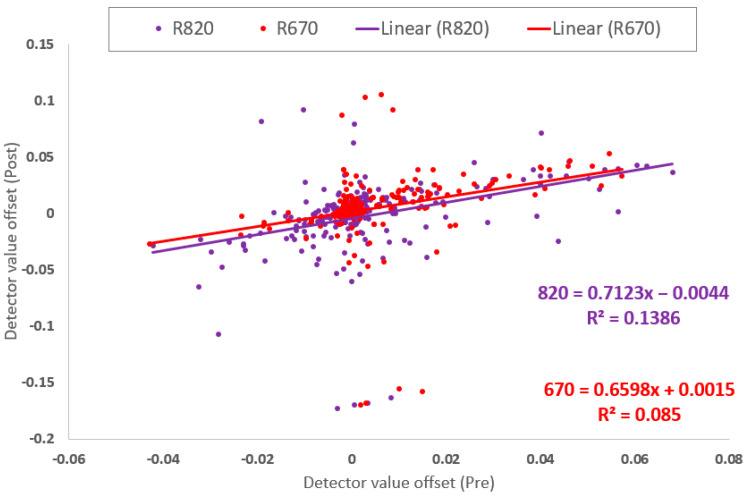
Chart of pre and post field data collections (years 2018–2021) sensor offset white panel measurements (for NIR and Red filtered detectors from sensors SN#201, SN#240, SN#241 and SN#252 mounted on the “Wolverine” proximal sensing cart). The divergence is shown from unity before and after each field data collection with the warmup and normalization process employed outside in full sunlight.

**Figure 11 sensors-23-05044-f011:**
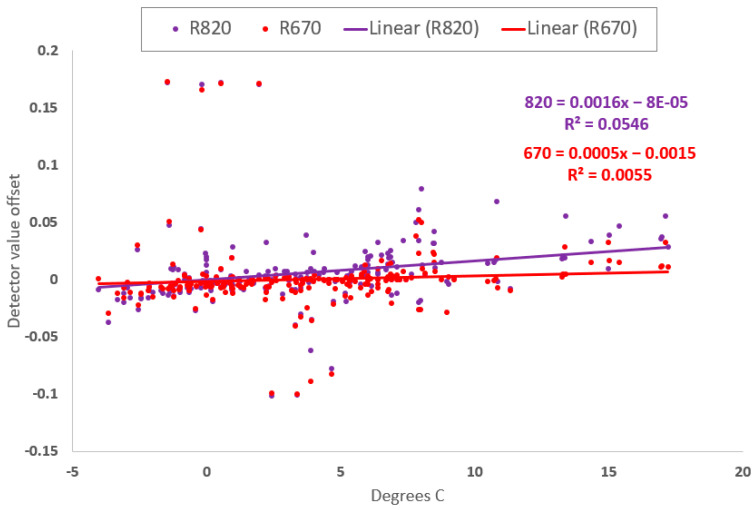
“Wolverine” proximal sensing cart years 2018–2021 field white panel summary data regression (for NIR and Red filtered detectors from sensors SN#201, SN#240, SN#241 and SN#252), temperature versus detector change where each data point is the median of one hundred data samples (last and first 20 s respectively) before and after field collection events from the normalized sensor white panel reference reflectance values.

**Table 1 sensors-23-05044-t001:** A summary of detector signal characteristics and reduction dependent on filter status, for unfiltered and filtered Red (670) and NIR (800) white panel measurements over 100 s (1 Hz samples) across the three SN#124 sensor detectors (R1, R2 and R3).

	Detector	Median	Maximum	Minimum	Standard Deviation
**Unfiltered Signal**	R1	0.98880	0.98890	0.98870	0.00005
	R2	0.98890	0.98900	0.98880	0.00005
	R3	0.98900	0.98910	0.98890	0.00005
**670 no normalization**	R1	0.04920	0.04940	0.04900	0.00011
	R2	0.04010	0.04040	0.03990	0.00009
	R3	0.04630	0.04670	0.04610	0.00011
**670 normalized**	R1	1.01265	1.01770	1.00610	0.00234
	R2	1.01090	1.01990	1.00320	0.00294
	R3	1.00405	1.01020	0.99880	0.00219
**800 normalized**	R1	1.17520	1.28110	1.06910	0.04018
	R2	1.05495	1.12790	1.00670	0.02218
	R3	1.02105	1.10920	0.92940	0.03860

## Data Availability

Related raw datasets are available from the USDA Ag Data Commons. https://data.nal.usda.gov/dataset/high-throughput-phenotyping-data-proximal-sensing-cart, accessed 28 January 2021, https://data.nal.usda.gov/dataset/bronson-files-dataset-1-field-17-2012, accessed 17 August 2021, https://data.nal.usda.gov/dataset/bronson-files-dataset-2-field-17-2013, accessed 3 September 2021, https://data.nal.usda.gov/dataset/bronson-files-dataset-3-field-107-2013, accessed 21 September 2021, https://data.nal.usda.gov/dataset/bronson-files-dataset-4-field-105-2013, accessed 30 September 2021, https://data.nal.usda.gov/dataset/bronson-files-dataset-5-field-105-2014, accessed 22 October 2021, https://data.nal.usda.gov/dataset/bronson-files-dataset-6-field-13-2014, accessed 2 December 2021, https://data.nal.usda.gov/dataset/bronson-files-dataset-7-field-13-2015, accessed 8 December 2021, https://data.nal.usda.gov/dataset/bronson-files-dataset-8-field-113-2016, accessed 31 March 2022, https://data.nal.usda.gov/dataset/bronson-files-dataset-9-field-113-2017-cotton, https://data.nal.usda.gov/dataset/bronson-files-dataset-10-field-113-2018-cotton, accessed 10 May 2022. Additional data is available upon reasonable request.

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
