# Peer review of "Proximal Active Optical Sensing Operational Improvement for Research Using the CropCircle ACS-470, Implications for Measurement of Normalized Difference Vegetation Index (NDVI)"

_sensors, 2023, doi:10.3390/s23115044_

Round 1
Reviewer 1 Report
Dear Authors,
in my opinion the uncertainty of measurement must be also presented especially when the signal quality is improved only by 3%. In introduction section you should also add the information which are given in datasheet by the manufacturer of ACS-470 about this unite. Maybe there is presented the influence of temperature on sensors behavior so the users are aware about this? If this information is not provieded in datasheet you should mention about it.
Author Response
Dear Reviewer, thank you for the constructive feedback and guidance in support of this article. I appreciate this learning experience and the challenge presented. I am honored to be part of this process and have worked hard to add value to the discussion on this topic. My hope is that I have sufficiently understood and addressed each point offered and that the community will benefit because of this review. The paper has been substantially revised for structure, clarity, and to add missing elements. I am grateful for your time and open to new learning going forward.
Point 1 - in my opinion the uncertainty of measurement must be also presented especially when the signal quality is improved only by 3%.
Response 1 - To support the minor improvement potential presented. I added more text to contextualize the expected small improvement as a guideline, and to underline how researchers should determine the performance of their equipment independently to derive a true correction. The improvement estimation was reworked to a median of 0.41% per C, using data collected before and after field events from 2018 – 2021, with an intention to be more relevant to future field research applications. (Originally the estimation was based on the initial sensor warming periods.) A STD of 3.13% was calculated, as well as an STE of 0.13%, and a 0.95 confidence interval of 0.26%. I also added two supplemental files “ANOVA.doc” which contains one-way ANOVA and HSD results for the data included in Figure 11, and “Bivariate.doc” which contains additional fit information on the detector by temperature change included in Figure 12.
Point 2 - In introduction section you should also add the information which are given in datasheet by the manufacturer of ACS-470 about this unite. Maybe there is presented the influence of temperature on sensors behavior, so the users are aware about this? If this information is not provided in datasheet you should mention about it.
Response 2 - The product manual was reviewed, and new language was added to describe the beam angle, and to detail that temperature influence was not originally described (https://manualzz.com/doc/7296492/crop-circle-handheld-system-operator-s-manual).
Reviewer 2 Report
The manuscript investigates the temperature sensitivity of silicone diode-based active radiometric reflectance sensing in high-throughput plant phenotyping (HTPP). It characterizes a customizable proximal active reflectance sensor's performance under temperature changes and field conditions, using titanium-dioxide white panels for normalization. The research offers recommendations for improving ACS-470 data quality based on sensor control data and field phenotyping experience.The research content of this paper is interesting, and the results have a certain reporting value. The text has a fair amount of length and an acceptable structure. Nonetheless, there are still some issues in this manuscript that need to be seriously addressed. For these reasons, I can only agree to publish this manuscript on Nanomaterials after major revisions.
The following issues should be appropriately improved and further elaboration:
1. The methodology section lacks a detailed explanation of the specific sensor configurations and environments tested in this study. The authors should provide a comprehensive description of the various setups, including sensor heights, angles, and ambient conditions, to provide a clear understanding of the experimental design.
2. The authors mention the use of an expanded user protocol involving (TiO2 painted) white normalization reference panels and sensor temperature tracking or control. However, they do not provide sufficient information on the actual implementation of these techniques in their experiments. The authors should elaborate on the materials, equipment, and procedures involved in using these normalization and temperature control methods.
3. In the results section, the authors claim an average improvement of 3% in raw ACS-470 detector signal quality using their enhanced protocol. However, they do not provide sufficient statistical evidence to support this claim. The authors should include the specific statistical tests used, along with their corresponding p-values, effect sizes, and confidence intervals, to strengthen the reliability of their findings.
4. The discussion section should be restructured to focus on the key findings and their implications in the context of the existing literature. The authors should critically compare their results with those from previous studies, highlighting the significance of their findings for agricultural research.
Minor editing of English language required
Author Response
Dear Reviewer, thank you for the constructive feedback and guidance in support of this article. I appreciate this learning experience and the challenge presented. I am honored to be part of this process and have worked hard to add value to the discussion on this topic. My hope is that I have sufficiently understood and addressed each point offered and that the community will benefit because of this review. The paper has been substantially revised for structure, clarity, and to add missing elements. I am grateful for your time and open to new learning going forward.
Point 1 - The manuscript investigates the temperature sensitivity of silicone diode-based active radiometric reflectance sensing in high-throughput plant phenotyping (HTPP). It characterizes a customizable proximal active reflectance sensor's performance under temperature changes and field conditions, using titanium-dioxide white panels for normalization. The research offers recommendations for improving ACS-470 data quality based on sensor control data and field phenotyping experience. The research content of this paper is interesting, and the results have a certain reporting value. The text has a fair amount of length and an acceptable structure. Nonetheless, there are still some issues in this manuscript that need to be seriously addressed. For these reasons, I can only agree to publish this manuscript on Nanomaterials after major revisions.
Response 1 - I revised text throughout the paper to reduce length and increase clarity where possible.
The following issues should be appropriately improved and further elaboration:
Point 2 - The methodology section lacks a detailed explanation of the specific sensor configurations and environments tested in this study. The authors should provide a comprehensive description of the various setups, including sensor heights, angles, and ambient conditions, to provide a clear understanding of the experimental design.
Response 2 - For the methodology section, more physical details describing the measurement setups and configurations was added, including specific naming of temperatures, distances, and angles. I worked to add more description to the design components to clarify exactly how and what was conducted.
Point 3 - The authors mention the use of an expanded user protocol involving (TiO2 painted) white normalization reference panels and sensor temperature tracking or control. However, they do not provide sufficient information on the actual implementation of these techniques in their experiments. The authors should elaborate on the materials, equipment, and procedures involved in using these normalization and temperature control methods.
Response 3 - Regarding the white panels, more information was added to better describe implementation of the different techniques and the reasons for the approaches used. Lines 154-160 describe materials used for the panels. Lines 204-210 address general implementation lines 283 – 286 address use in the lab, lines 400-402 mention use outdoors, and citation 65 offers more explanation. Lines 413 – 422 describe how to evaluate data values in the white panel normalization process. Lines 498-512 describe how the panel can actually be used in experiments. Citation 70 examples how the process was used in another paper. Citations 72-74 contain additional description on how the process can be conducted and provide example data sets used in other publications.
Point 4 - In the results section, the authors claim an average improvement of 3% in raw ACS-470 detector signal quality using their enhanced protocol. However, they do not provide sufficient statistical evidence to support this claim. The authors should include the specific statistical tests used, along with their corresponding p-values, effect sizes, and confidence intervals, to strengthen the reliability of their findings.
Response 4 - The improvement estimation was reworked to a median 0.4% per C using data collected before and after field events from 2018 – 2021, with an intention to be more relevant to future field research applications. (Originally the estimation was based on the initial sensor warming periods.) More statistical information was added to support the results, based on the 2018 to 2021 control data. A STD of 3.13% was calculated, as well as an STE of 0.13%, and a 0.95 confidence interval of 0.26%. I also added two supplemental files “ANOVA.doc” which contains one-way ANOVA and HSD results for the data included in Figure 11, and “Bivariate.doc” which contains additional fit information on the detector by temperature change included in Figure 12. Language was also added to underline that each detector behaves differently, so an individual white panel measurement before and after every field collection is needed to truly determine the correction potential, and the expected improvement is offered only as a generalized estimate to contextualize the phenomena.
Point 5 - The discussion section should be restructured to focus on the key findings and their implications in the context of the existing literature. The authors should critically compare their results with those from previous studies, highlighting the significance of their findings for agricultural research.
Response 5 - The discussion was revised and restructured with intention to focus on findings and implications. Additional language was added to underline the key finding that some quality improvement may be possible for this sensor when white panels etc. are used. Additional language was added to mention that hours of white panel pre / post field collection data has not been previously reported for the ACS-470, so it is hard to make direct comparisons with existing literature; and that there is a unique quality with the sensor being proprietary active technology and customizable. This makes a clear evaluation with other work difficult, however, additional language was added to explain the context of this sensor within the general agricultural research reflectance measurement category. Language was included to detail specific sensor differences, such as regarding the GreenSeeker with citation 16, and to highlight areas of consideration for researchers such as the bandwidth and centering of the multi-spec reflectance used and how a filter would optimally be positioned on an ASC-470 in order to expand the research utility of the product from its original management orientation to a possible support in crop breeding or simulation modeling. Implications of the fact that temperature change alone did not explain all the detector value change measured and that there were a few instances of irregular data points, were included to support use of the white panel reference. Also, the theoretical effect that detector drift would have on an NDVI calculation was included as an implication. A paragraph was added at the end of the discussion to summarize the practical application of the process.
Reviewer 3 Report
The experimental work presented in the Manuscript, entitled „ Proximal Active Optical Sensing operational improvement for research using the CropCircle ACS-470, implications for measurement of normalized difference vegetation index (NDVI) ". The purpose of this study was to characterize the only currently customizable proximal active reflectance sensor available for HTPP research under changes in temperature and in field conditions, and to suggest an operational use approach for researchers. Sensor performance was measured using large titanium-dioxide white painted field normalization reference panels and the consequent sensor detector values as well as sensor body temperatures were recorded.
Abstract
· Abstract is more description and it does not support by enough digital results.
· Last sentence should be improved to reflect the conclusion of the abstract.
Introduction
· Please, highlight what is the novelty (originality) of the work because other studies presented similar to this work? And what is new in your work that makes a difference in the body of knowledge?
· Please present the previous studies in the introduction?
Materials and methods, results and discussion are well written
· Figure 5 should be improved
· Please, write the practical applications of your work in a separate section, before the conclusions and provide your good perspectives.
· Please write about the limitations of this work in details in conclusion section.
Minor editing of English language required
Author Response
Dear Reviewer, thank you for the constructive feedback and guidance in support of this article. I appreciate this learning experience and the challenge presented. I am honored to be part of this process and have worked hard to add value to the discussion on this topic. My hope is that I have sufficiently understood and addressed each point offered and that the community will benefit because of this review. The paper has been substantially revised for structure, clarity, and to add missing elements. I am grateful for your time and open to new learning going forward.
The experimental work presented in the Manuscript, entitled „ Proximal Active Optical Sensing operational improvement for research using the CropCircle ACS-470, implications for measurement of normalized difference vegetation index (NDVI) ". The purpose of this study was to characterize the only currently customizable proximal active reflectance sensor available for HTPP research under changes in temperature and in field conditions, and to suggest an operational use approach for researchers. Sensor performance was measured using large titanium-dioxide white painted field normalization reference panels and the consequent sensor detector values as well as sensor body temperatures were recorded.
Abstract
Point 1- Abstract is more description and it does not support by enough digital results.
Response 1 - The abstract was revised, and more quantitative results were added to the abstract including a temperature, a height, and a signal value. Statistics were also added to the abstract to better describe the typical 0.24% drift measured from pre to post field conditions where the sensor temperature changed more than 1 C.
Point 2 - Last sentence should be improved to reflect the conclusion of the abstract.
Response 2 - The last sentence of the abstract was re-written to better describe what the paper is trying to present and from where the findings are developed. More important than the specific results of how our detector values changed with temperature, is the idea that a researcher may use sensor temperature tracking and insulation to mitigate thermal effects and use the white panel reference to characterize detectors and possibly correct field measurements in their own design.
Introduction
Point 3 - Please, highlight what is the novelty (originality) of the work because other studies presented similar to this work? And what is new in your work that makes a difference in the body of knowledge?
Response 3 - The novelty of the work was better expressed to explain that years of ACS-470 sensor white panel experimental control data before and after field collections and their implications has apparently not been reported previously. Likewise, the minor behavior of the unique active proprietary custom filtered detectors on a white panel has not previously been reported. Other studies typically compare the ACS-470 to different sensors or imagery, where what is new here is that many ACS-470 sensors were compared to each other with temperature tracking over years using the reference panel and where a temperature influence was seen. However more importantly, it was shown that detectors each behave differently, so a researcher could be supported by using the white panel to quantify their own customized ACS-470 setup, and a minor increase in reflectance data quality control may empower higher-level research such as breeding selections or crop simulation modeling. We have also never seen a reporting of the “raw” detector signal measured with the filters removed and over a temperature change to show temperature influence and signal quality of the full spectral input at once. Finally, the wide body of technical experience using the sensing technology in many experiments allowed this paper to present different illustrative examples of scenarios related to research but using the same or related sensor units as tracked by serial number, and to provide detailed operational descriptions and suggestions for researchers, including the final generalized 2.7% expected improvement potential.
One other aspect of relevancy is that although this paper focuses on daytime nadir-view NDIV, the custom ability of different color filters to be used and normalized in the ACS-470 means that this product is the only avenue to measure active blue, green, or yellow wavelengths and it allows sensing at angles and nighttime conditions. Information for red-edge, green and yellow filtered detectors is presented in the appendix and supplemental information.
Point 4 - Please present the previous studies in the introduction.
Response 4 - Two additional studies were added to the introduction 16 and 17. These papers present the GreenSeeker which is another active optical NDVI technology (although not customizable) and provide a modern review of the phenotyping arena. Other studies are presented that specifically focus on ACS-470 usage such as 36 – 50, and the use of the ACS-470 sensor in experiments 65, 66, 68-72, 74, and 80.
Point 5 - Materials and methods, results and discussion are well written
Response 5 - Thank you kindly
Point 6 - Figure 5 should be improved
Response 6 - The figure was replaced with a table to improve clarity.
Point 7 - Please, write the practical applications of your work in a separate section, before the conclusions and provide your good perspectives.
Response 7 - More description on practical applications for researchers was added to the discussion, such as how the sensor can be reliably utilized and what specific issues to watch out for. A description of how to address data quality improvement is included. An example of research employing some of these ideas was offered in citation 68, and a description of the normalization process is given starting on line 496. The practical application of temperature measurement starts on line 513, and for NDVI determination on line 533 where possible implications of detector drift are explained. Additional text was added at line 548 to summarize what specifically is the primary practical application. The citations 65, 66, 68-70 and 74 offer examples where the process was applied in other research. Also, there is a preliminary description of how the white panel results are used as a method on lines 211-215.
Point 8 - Please write about the limitations of this work in details in conclusion section.
Response 8 - A paragraph that speaks to the limitation of the work was added at lines 591-603.
Reviewer 4 Report
First of all, I want to congratulate the authors for their efforts in this manuscript. This manuscript proposes a proximal sensing based on active optical to measure the NDVI for controlling crop status. In general terms, the paper is well written, and the provided results analyses are very good. Nevertheless, there are some aspects to be improved. Following, I include a list of aspects to be improved in order to enhance the quality of the manuscript.
Major issues:
The introduction is excessively complex. I suggest moving the content related to the literature review to a new section placed after the introduction in which the related work is presented.
After moving the related work content to the new section, the introduction should be extended, focusing on the topic of the paper. Three new paragraphs should be added detailing the aim and the structure of the paper. For the first paragraph, general details of plant monitoring approaches and differences between proximal and remote monitoring should be provided with regards to NDVI and phenotyping (consider this paper for the contextualization: https://doi.org/10.1016/j.agwat.2022.107581 and https://doi.org/10.1016/j.xplc.2022.100344)
In the second paragraph, the aim of the paper as well as the main novelty should be described; consider using bullet points. The third paragraph, in which the structure of the paper is described, should be included after the aim of the paper.
Consider dividing the Material and Methods section into different subsections describing the different aspects of the proposal and the methodology followed.
In the result section, the authors should include a short paragraph at the beginning describing the content and order of the analysis of results. If possible, consider adding different subsections for the different aspects included in this section.
In the discussion, a comparison with existing techniques should be provided. The authors have to consider the use of a Table to summarize the content.
In the conclusions, add the future work in a new paragraph.
General comments:
Consider avoiding using the same terms which are already used in the title.
Check the captions of Figures and reduce them if possible. The current extension of Figure captions difficulties their read.
Line: 429: Use an equation for this type of content.
Author Response
Dear Reviewer, thank you for the constructive feedback and guidance in support of this article. I appreciate this learning experience and the challenge presented. I am honored to be part of this process and have worked hard to add value to the discussion on this topic. My hope is that I have sufficiently understood and addressed each point offered and that the community will benefit because of this review. The paper has been substantially revised for structure, clarity, and to add missing elements. I am grateful for your time and open to new learning going forward.
First of all, I want to congratulate the authors for their efforts in this manuscript. This manuscript proposes a proximal sensing based on active optical to measure the NDVI for controlling crop status. In general terms, the paper is well written, and the provided results analyses are very good. Nevertheless, there are some aspects to be improved. Following, I include a list of aspects to be improved in order to enhance the quality of the manuscript.
Major issues:
Point 1 - The introduction is excessively complex. I suggest moving the content related to the literature review to a new section placed after the introduction in which the related work is presented.
Response 1 - The ACS-470 related primary literature review was moved several paragraphs after an introductory description of the novelty and structure of the paper. The introduction was revised overall to clarify and reduce complexity while still trying to cover the breath of NDVI measurement. Detail was included on what makes the active ACS-470 sensing unique from other typical remote sensed or proximal passive and the GreenSeeker active measurement. My intention was to be comprehensive enough to describe the reflectance category as it applies to agricultural research but also detail some of the seldom mentioned nuance, so that a researcher could glean a minor detail or general understanding.
Point 2 - After moving the related work content to the new section, the introduction should be extended, focusing on the topic of the paper. Three new paragraphs should be added detailing the aim and the structure of the paper. For the first paragraph, general details of plant monitoring approaches and differences between proximal and remote monitoring should be provided with regards to NDVI and phenotyping (consider this paper for the contextualization: https://doi.org/10.1016/j.agwat.2022.107581 and https://doi.org/10.1016/j.xplc.2022.100344)
Response 2 - The introduction was extended, firstly to add a paragraph that addresses general details of plant monitoring approaches. The suggested papers were included to provide more context around NDVI and phenotyping and provide relevancy for data quality.
Point 3 - In the second paragraph, the aim of the paper as well as the main novelty should be described; consider using bullet points. The third paragraph, in which the structure of the paper is described, should be included after the aim of the paper.
Response 3 - A paragraph regarding the aim and novelty of this work was added. The benefit of using a white panel to resolve detector performance and its implications is introduced. The novelty of presenting white panel reflectance data specific to the unique ACS-470 that also includes temperature tracking is presented.
A paragraph describing the structure of the paper was added. A brief mention of the contents for each section is offered, to provide a descriptive outline of what will be presented.
Topic focused bullet points were considered, but a description of the aim and novelty of the paper was selected instead, due to the somewhat wide range in topics and their implications. For instance, the aim is generally to describe several minor signal quality features that may have been invisible to a research user because they are sometimes transient and influenced by many different factors, while the novelty stems more from the large body of experience formed over many years and experiments and the lack of other reporting. I wasn’t sure how to better list all these items absent their associated relations and details, and if I included those item details then it seemed the narration would become lengthy. Hopefully the aim and novelty are now clearly expressed in the introductory paragraph three, and the structure delineated in paragraph four.
Point 4 - Consider dividing the Material and Methods section into different subsections describing the different aspects of the proposal and the methodology followed.
Response 4 - The materials and methods section was divided into seven paragraphs for clarity. Each paragraph was reviewed with the intention to better describe the aspect or rationale of each proposed approach, and then how the method conducted aimed to meet that target. Furthermore, bulleted headings could be added as sub-section elements if allowed by the editors (Equipment, Sensor reflectance, Controlled environment, Field conditions, Method of use, Supporting data metrics and data processing).
Point 5 - In the result section, the authors should include a short paragraph at the beginning describing the content and order of the analysis of results. If possible, consider adding different subsections for the different aspects included in this section.
Response 5 - A new paragraph was added to the start of the results section to describe the order of the results content. Subsections were included for sensor temperature influence on filtered and unfiltered detectors, the influence of filters on the data signal, influence of environmental conditions regarding temperature, mitigation of environmental temperature influence, and signal response outdoors during warmup and in relation to temperature, as well as the summary pre / post field collection control data results.
Point 6 - In the discussion, a comparison with existing techniques should be provided. The authors have to consider the use of a Table to summarize the content.
Response 6 - A Table was added to replace figure 5, and a summary list of the techniques was included at the end of the discussion. Authors are unaware of other studies which evaluate the ACS-470 signal quality on a white panel using temperature tracking to characterize individual detector behavior and measure detectors before and after field collections as a basis to report possible data corrections based on signal divergence from unity. The other techniques of reflectance sensing seemed to be outside the scope of the paper as they are quite varied and nuanced.
Point 7 - In the conclusions, add the future work in a new paragraph.
Response 7 - A future work paragraph was added to the conclusions section lines 579-590. The suggestion that further data quality improvement could be attained using active thermal control on the sensor is presented along with a basic description and citations (79, 80) which relate to the approach. A single 60-watt TEC chip with PC style atmospheric heat sink was attached to one ACS-470 and signal quality measured in the lab but the power was insufficient to control the sensor temperature, so this was not presented.
General comments:
Point 8 - Consider avoiding using the same terms which are already used in the title.
Response 8 - The paper was revised, looking to make descriptive terms more unique and to not repeat terms whenever possible.
Point 9 - Check the captions of Figures and reduce them if possible. The current extension of Figure captions difficulties their read.
Response 9 - Figure captions were revised, and they were reduced in several places to support a more clear and concise description and improve readability.
Point 10 - Line: 429: Use an equation for this type of content.
Response 10 - The text line was replaced with an equation, and other instances were also changed, perhaps there is a better graphical representation that journal editors can apply.
Round 2
Reviewer 1 Report
Dear Authors,
Thank you for revising the manuscript and for the responses.
Kind regards,
Reviewer 4 Report
I suggest the authors use subsections instead of bullet points for the results and include the subsections in section 2. Thus, the structure of the paper will be more "reader-friendly".